# Silicone Nanocomposites with Enhanced Thermal Resistance: A Short Review

**DOI:** 10.3390/ma17092016

**Published:** 2024-04-25

**Authors:** Maria Zielecka, Anna Rabajczyk

**Affiliations:** Scientific and Research Centre for Fire Protection-National Research Institute, Nadwiślańska 213, 05-420 Józefów, Poland; arabajczyk@cnbop.pl

**Keywords:** silicone nanocomposites, thermal stability, nanoadditives

## Abstract

Continuous technological progress places significant demands on the materials used in increasingly modern devices. An important parameter is often the long-term thermal resistance of the material. The use of heat-resistant polymer materials worked well in technologically advanced products. An economically justified direction in searching for new materials is the area of polymer nanocomposite materials. It is necessary to appropriately select both the polymer matrix and the nanofillers best able to demonstrate the synergistic effect. A promising area of exploration for such nanocomposites is the use of organosilicon polymers, which results from the unique properties of these polymers related to their structure. This review presents the results of the analysis of the most important literature reports regarding organosilicon polymer nanocomposites with increased thermal resistance. Particular attention was paid to modification methods of silicone nanocomposites, focusing on increasing their thermal resistance related to the modification of siloxane molecular structure and by making nanocomposites using inorganic additives and carbon nanomaterials. Attention was also paid to such important issues as the influence of the dispersion of additives in the polymer matrix on the thermal resistance of silicone nanocomposites and the possibility of modifying the polymer matrix and permanently introducing nanofillers thanks to the presence of various reactive groups. The thermal stability mechanism of these nanocomposites was also analysed.

## 1. Introduction

The continuous development of modern devices in various industrial areas, from construction to advanced electronics, places increasing demands on materials, including polymer materials used in innovative technologies. The long-term thermal resistance of a material is often an important parameter influencing the possibility of its use in technologically advanced products [1,2]. The use of heat-resistant polymer materials has worked well in many products. However, with the development of industrial technologies, the difference between the thermal stability of the polymer matrix and industrial requirements is gradually increasing, which necessitates the search for new material solutions.

One of the effective and economically justified directions in the search for new materials is in the area of polymer nanocomposite materials, which have been widely developing in recent years. Wide possibilities of obtaining the desired properties of these nanomaterials can be achieved thanks to the optimal selection of both the polymer matrix and nanofillers. Organosilicon polymers, commonly called silicones due to their unique chemical and physical properties resulting from the rich silicon chemistry, are often the first-choice material when designing polymer nanocomposite materials for demanding innovative applications (Figure 1).

The main types of organosilicon polymers used to obtain nanocomposites are polymers with a branched structure—silicone resins and those with a linear structure—silicone rubbers. The structure of organosilicon polymers differs significantly from the structure of organic polymers, especially in terms of interatomic distances, angles and energy of bond formation [3,4], as illustrated by the data in Table 1.

Moreover, the partially ionic nature of the Si-O bond related to the formation of a dπ-pπ bond between Si and O atoms has a significant impact on the much better thermal resistance of silicon–organic polymers, compared to conventional organic polymers. This nature of the bond formed significantly increases the dissociation energy of the Si-O bond, which is 108 kcal/mol compared to 85.2 kcal/mol for the C-C bond and 82.6 kcal/mol for the C-O bond [5]. This results in a significant increase in the stability of the Si-O bond at elevated temperatures, which translates into increased heat resistance of organosilicon polymers compared to organic polymers. Organic substituents on silicon atoms also have a significant impact on the heat resistance of organosilicon polymers. They can be arranged as follows, with decreasing thermal stability: C_6_H_5_ > ClC_6_H_4_ > Cl_3_C_6_H_2_ > Cl_2_C_6_H_3_ > CH_2_ = CH > CH_3_ > C_2_H_5_ [6].

The aim of this review is to present the most important information regarding silicone nanocomposites with increased thermal resistance based on a literature review, including the most important patents. The overview will discuss modification methods of silicone nanocomposites, focusing on increasing their thermal resistance related to the modification of siloxane molecular structure, and by making nanocomposites using inorganic additives and carbon nanomaterials. In addition, the impact of dispersing additives in the polymer matrix on the thermal resistance of silicone nanocomposites will be analysed, as well as the possibility of modifying the polymer matrix and permanently introducing nanofillers due to the presence of various reactive groups. The mechanism of thermal stability of silicone nanocomposites will be also analysed. The literature review was performed using the following databases: Web of Knowledge, Scopus, and Google Scholar, taking into account the following keywords: silicone nanocomposites, thermal stability, nanoadditives, and application. The research also covered Espacenet, Patentscope, and Google Patents, taking into account the years 2015–2024, which resulted in the presentation of selected patents relevant to the subject. We used the keywords: silicone nanocomposites, thermal stability, nanoadditives, i.e., appropriately to the article. For example, as a result of the Google patents search, we received 28 results. However, the vast majority of patents focused on mechanical properties, which did not fit the scope of the presented article. This review does not cover the use of flame retardants, especially halogenated ones. The use of such additives is limited in European Union countries and the USA on the basis of legal provisions (e.g., REACH regulation, RoHS Directive).

## 2. Modification Methods of Thermal Stability of Silicone Nanocomposites

### 2.1. Enhancement of Thermal Stability by Modification of Siloxane Molecular Structure

The rich chemistry of inorganic and organic silicon compounds creates various possibilities for obtaining organosilicon polymers with different structures. Taking into account the structure of the polysiloxane chain, the following basic types of organosilicon polymers used in modern products can be distinguished: silicone oils and rubbers with a linear structure and silicone resins with a branched structure. Depending on the degree of branching, organosilicon polymers show differences in thermal stability. The thermal degradation processes of polysiloxanes occurring at elevated temperature and in the presence of oxygen include the following two basic phenomena:-Breaking the Si-O-Si bond leading to a rearrangement reaction;-Oxidation of side groups in the polysiloxane chain causing a change in the structure of the polysiloxane chain.

Based on thermogravimetric tests of linear and branched organosilicon polymers, the thermal stability of branched polymers was found to be significantly higher than that of linear polymers [7]. The measure of the degree of branching of organosilicon polymers is the R/Si ratio, which determines the ratio of organic groups to silicon atoms, which decreases with the increase in the degree of branching. The lower the R/Si ratio, the greater the T unit content and the degree of branching. Obtaining the desired R/Si is possible by appropriate selection of linear or branched monomers for the synthesis of the organosilicon polymer [8]. The detailed effects of R/Si and Ph/Me on thermal stability based on T_5_ measured by the temperature at 5% weight loss in the TGA curve are shown in Table 2.

The characteristics of polysiloxanes with a degree of branching R/Si ranging from 1.2 to 2, and a molar ratio of the content of phenyl and methyl groups ranging from 0/100 to 100/0 mol/mol% (given in Table 2), indicate a significant impact of these parameters on thermal stability. The solid residue during the decomposition of branched polysiloxanes at a temperature of 800 °C was, depending on the content of phenyl groups, 66.5–40.5 wt.% in the air atmosphere. At the same time, it can be concluded that the amount of solid residue during the decomposition of polysiloxanes increases with the increased content of methyl groups. Linear polysiloxane with a branching degree of 2 and a phenyl and methyl group content ratio of 75/25 was characterized by a constant residue of 37.2 wt.% and 26.4 wt.% in the air atmosphere. The value of solid residue after thermal decomposition is a good, but not the only, criterion for thermal stability. An important parameter, especially from the point of view of application possibilities, is the degradation temperature, which increases with the content of phenyl groups and the degree of branching.

The relationships discussed above can also be clearly observed when analysing the properties of highly branched MQ silicone resins consisting of a monofunctional siloxane (R_3_SiO_1/2_, the M units) and a tetrafunctional siloxane (SiO_4/2_, the Q units) [15,16]. These polymers have favourable thermal stability due to the formation of new regular macromolecular structures and the introduction of phenyl groups [17]. These inorganic/organic hybrid materials can be used as a potential component of temperature-resistant electronics adhesives, heat-resistant coatings, and liquid silicone rubber. Based on the analysis of the influence of M/Q values calculated from the total carbon and hydrogen content in the resin, it was found that an increase in the amount of the Q segment (SiO_4/2_) improves the thermal stability of MQ resins [18]. Moreover, the thermal stability of composites of some polysiloxanes, e.g., silicone rubber, can be influenced by the selection of the type and place of incorporation and the number of active groups in the polysiloxane molecular chain [19].

Moreover, it should be emphasized that, in the processes of thermal decomposition of polysiloxanes, especially at low temperatures, impurities that catalyse the degradation process may play a significant role. Acidic or alkaline pollutants [20], oxygen [21,22] and water have a significant impact, and can also influence the degradation rate. Degradation may also occur due to changes in the interactions between the polymer and the reinforcing silica filler [23,24].

### 2.2. Enhancement of Thermal Stability by Inorganic Nanoadditives

An effective method to improve the thermal stability of polysiloxane composites is to introduce inorganic additives into their structure. Additives such as iron octanoate, fumed silica and benzoyl peroxide as a cross-linking agent enable improved thermal stability [25]. The extensive development of the possibility of using inorganic nanoadditives has enabled a significant improvement in the thermal stability of siloxane composites by limiting the migration of polysiloxane chains [26], creating a spatial heat conduction network by nanoadditives [27] and increasing the degree of dispersion of nanoadditives while reducing their quantity [28]. Nanoparticles of oxygenated aluminium compounds are a widely used additive to organosilicon polymer composites that improve their thermal stability. However, achieving the desired effect is usually possible by using such additives in an amount of 20–30% by weight, which is unfavourable from the point of view of deteriorating some other properties of the composites, especially their strength. However, the use of nano Al_2_O_3_ in an amount from 1 to 5% by weight, in nanocomposites of organosilicon polymers used as insulators, allows for obtaining very good thermal stability [29]. It was found that the best thermal and dielectric properties of the nanocomposite were obtained by introducing 3 wt.% nano Al_2_O_3_.

Al_2_O_3_ nanoparticles were also used by Lorenzo [30]. Aluminium oxide nanoparticles were combined with SiO_2_ nanoparticles to obtain the SiO_2_@Al_2_O_3_ nanofiller, which was used to fill SBR–BR nanocomposites. An improvement in thermal transport properties of ~30% was observed compared to the reference composite containing only 60 phr silica, and ~80% for the pure polymer blend [30].

A significant improvement in the dielectric and thermal properties of organosilicon polymer nanocomposites was also achieved when using nano boron nitride (BN) in a mixture with nano Al_2_O_3_ [31]. The influence of the size and shape of micro (particle size in the range of 7.5 to 23.8 µm) and nano (particle size 70 nm, with partial particle aggregation) boron nitride particles on the properties of organosilicon polymer composites was investigated [32,33]. The tensile strength of such composites with the addition of any type of boron nitride was found to deteriorate, which means that the interfacial interaction between the organosilicon polymer and boron nitride is weak. Nano-size fillers have a more pronounced effect on the modulus and tensile strength compared to micron-size boron nitride. It was found that the effect of the filler aspect ratio is very critical to obtain high thermal conductivity. The highest thermal conductivity of all five types of BN is provided by boron nitride containing plate-shaped particles [32,33].

The work of Farahani et al. [34] indicated the possibility of using boron nitride nanosheets (BNNS), including those obtained by exfoliation of hexagonal boron nitride (h-BN) using monoethylene glycol as an exfoliating agent. Nanosheets added to silicone rubber nanocomposites at 3 and 5 wt.% significantly increased the thermal conductivity of silicon rubber (SR) nanocomposites. Additionally, silicone rubber that contained 5 wt.% h-BN and exfoliated BN exhibited approximately 20-fold higher thermal conductivity compared to pure SR. However, it was found that h-BN exfoliation had a negligible effect on improving thermal properties [34].

A significant increase in thermal stability was also achieved by using layered Mg-Al hydroxide obtained by solution intercalation [35]. It was found that the temperature corresponding to a loss of 50% of the mass of a silicone rubber composite containing 1 wt.% layered Mg-Al hydroxide, was 20 °C higher compared to the temperature for the composite not containing this additive. This phenomenon could be attributed to the fact that the layered Mg-Al hydroxide dispersed in the polymer matrix reduced the polysiloxane chain breakage and inhibited the formation of volatile products.

In tests of the thermal stability of silicone nanocomposites, a clear influence of the additive content and their synergistic effect, as well as the interaction of the polymer matrix with the additives, was found. In a silicone nanocomposite containing SiO_2_ nanoparticles and micro aluminium nitride, it was found that nano-SiO_2_ acted as a bridge between the polysiloxane chains and micro aluminium nitride, increasing the contact area between the silicone rubber molecular chains and the additive molecules [36].

Moreover, the existence of hydrogen bonds between nanofillers and polysiloxane chains means the migration of chains at the phase boundary has been limited, which has a clear positive effect on the thermal stability of the nanocomposite [37].

However, the ability to modify the surface of nanofillers is important, a good example of which are halloysite nanotubes (HNTs) [38]. Thanks to the structure of these nanotubes, interactions with polysiloxane and modification of their surface are possible, which significantly improves the thermal properties of silicone nanocomposites [39]. HNTswere used to modify a polyetherimide/silicone rubber nanocomposite. The obtained nanocomposite was characterized by increased thermal stability, with maximum values obtained at an HNT loading of 3 phr. It was also found that the interfacial and intertubular interactions between HNTs and polymer matrices and the formation of zigzag structures of HNTs are the main reasons for the improvement of various properties [40].

Improved thermal stability was also achieved for nanocomposites of fluorosilicone rubber with CeO_2_ nanoparticles [41]. The study of the degradation mechanism of the fluorosilicone rubber composite showed that at temperatures below 350 °C, the most important factor influencing the degradation process was the oxidative cleavage of the side groups, while the degradation associated with the siloxane rearrangement was less important. Moreover, it was found that the addition of cerium oxide significantly improved the thermal stability of such composites by inhibiting oxidative fission. The influence of the cerium oxide structure was compared by examining the influence of cerium oxide with a surface-modified layered structure and cerium oxide nanoparticles with a particle size of 20–50 nm. Both additives had specific surface area and good dispersibility and showed significant antioxidant activity at 230 and 250 °C, respectively [41].

The modification was also carried out by using TiO_2_ nanoparticles, which allowed for obtaining a TiO_2_/silicone nanocomposite with high thermal stability (73 °C increase in weight loss by 50%). The titanate coupling agent-treated S-TiO_2_ nanoparticles were incorporated into the diluted silicone resin by simply mixing the solvent. Layers of titanate coupling agent formed on the surfaces of TiO_2_ nanoparticles with a diameter of 3–4 nm. This mechanism allows for excellent dispersion of S-TiO_2_ nanoparticles in the silicone resin and excellent compatibility between TiO_2_ nanoparticles and silicone resin [42].

Metal nanoparticles incorporated into the pores of porous silicon, through direct synthesis, allowed for the preparation of a nano-porous silicon–nickel nanocomposite (nPS/Ni) for thermal insulation applications. The nickel element is chemically deposited while the nanoparticles precipitate on the pore surface. The nanocomposites obtained in this way (nPS/Ni) show better thermal stability at up to 900 °C than materials without the addition of nickel (nPS) at a temperature of 600 °C [43].

A beneficial effect of ozone modification of the surface of nanoparticles was found in the case of nanosilica used as a component of a silicone rubber nanocomposite [44]. Surface modification of silica nanoparticles improved the dispersion of this nanofiller and increased the intermolecular interactions of polysiloxane-nanofiller, a result of which was that the degradation of silicone rubber in a high-temperature environment required more energy to overcome the interactions between the additive and the matrix. This improves the thermal stability of silicone rubber composites.

### 2.3. Enhancement of the Thermal Stability by Carbon Nanomaterials

In recent years, intensive research has been carried out on the use of carbon nanotubes and graphene as nanofillers in polymer composites, which is due to the unique properties of these nanomaterials. Because both carbon and graphene nanotubes aggregate easily, most research focuses on developing optimal methods for surface modification and introducing these nanomaterials into polymer nanocomposites. This allows them to be well dispersed in the structure of the polymer matrix. However, it should be added that there are many factors that determine this process, including: the presence of appropriate groups or the amount of nanoadditives (Figure 2).

Multi-walled carbon nanotubes modified with ethyl 4-aminocinnamate using the ultrasonic cavitation technique were used in an amount from 0.6 to 1 wt.% as a component of a silicone rubber nanocomposite. This enabled a significant improvement in the properties of the obtained silicone rubber, including thermal stability. The initial decomposition temperature of the composite without nanotubes was 217 °C, and for the nanocomposite containing 0.6, 0.8 and 1 wt.%, it was 319, 324 and 335 °C, respectively [45].

The influence of π-π coupling between carbon nanotubes (CNTs) and phenyl groups on the thermal stability of phenyl-silicone rubber nanocomposites was investigated. It was shown that as the content of phenyl groups in the polysiloxane chain of the polymer matrix increased, the interaction between the nanotubes and the phenyl group increased. The T5 value (temperature at 5% mass loss on the TGA curve) of silicone rubber with a high content of phenyl groups increased significantly. Improvement of the thermal stability of the silicone rubber nanocomposite was also achieved by introducing 0.05 wt.% carbon nanotubes in optimized conditions using the wet milling method [28]. It should be emphasized that the dispersibility of CNTs in their nanocomposites is a key factor in their impact on improving the thermal stability of nanocomposites.

Results confirming the relationship between the degree of dispersion of the nanofiller and the properties of the polymer nanocomposite were also obtained by examining nanocomposites of organosilicon polymers with graphene nanoribbons [46]. The influence of the degree of dispersion on the thermal stability of the silicone rubber nanocomposite was confirmed by X-ray analysis. Similarly, many researchers have wondered about the effect of graphene nanosheets on the thermal conductivity and thermal stability of silicone rubber [47]. When the graphene content was from 2 wt.% up to 8% by weight, the T_5_ of the nanocomposite remained at approximately 392 °C. However, for unmodified silicone rubber, this temperature was about 20 °C lower. Moreover, the study results also showed that the surface of graphene nanosheets contains many functional groups capable of forming bonds at the interface between graphene and silicone rubber. This system allowed for reducing the mobility of polysiloxane chains, which improved the thermal stability of graphene/silicone rubber nanocomposites.

The influence of graphite oxide produced from expanded graphite on the thermal stability of graphite oxide/silicone rubber nanocomposites obtained by solution intercalation was also investigated [48]. Thermal stability tests were carried out using the thermogravimetric method, both in air and in a nitrogen atmosphere. When the graphite oxide content was 0.6 wt.%, the temperature at 10% weight loss in the TGA curve in air or nitrogen atmosphere was 410.5 and 421 °C, respectively. This represented an insignificant increase of 9.6 and 15 °C, respectively, compared to unmodified silicone rubber. However, the above increase in thermal stability is not significant, but can be attributed to the interaction of -OH and -COOH in graphite oxide with a polar -Si-O- bond. This interaction increased the stiffness of the siloxane chain, making degradation of the polysiloxane chain more difficult [48].

An interesting method is the use of γ radiation to polymerize 3-methacryloxypropyltrimethoxysilane (MPTMS) on the surface of graphene oxide (GO) and carbon nanotubes (CNT). The fabricated GO/CNT-Si not only showed better dispersion, but also improved the SR mechanical properties, as well as thermal properties, including the thermal expansion coefficient, thermal stability and conductivity [49].

Silicone rubber (SR)/vinyl graphene oxide (vinyl-GO) nanocomposites are also characterized by higher thermal stability compared to pure SR. This nanocomposite was obtained through the hydrosilylation reaction of hydrogen polydimethylsiloxane silicon (H-PDMS) with polyvinyl dimethylsiloxane (vinyl-PDMS), with vinyl-GO used as a nanofiller [50], whereas Wang et al. [51] modified carbon fibres (CFs) by grafting nanodiamond (ND) particles onto their surface. Then, the nanofillers prepared in this way were used to obtain a CF-ND/silicone rubber (SR) composite. ND acted as a “bridge” between CFs. When the CF-ND content (1:6) was 20%, the thermal conductivity of the SR composite was 69% higher than that of pure SR. The CF-ND/SR composites also showed excellent thermal stability [51].

Lorenzo [30] also noticed the benefits of introducing a filler in the form of graphite nanoparticles, as they allow for improving the thermal properties of polymers. The deposition of silica on the surface of expanded graphite (EG) or graphite nanoparticles modified with silica nanoparticles (EG@SiO_2_) provides improved thermal stability and, at 1000 °C, a residual mass (~18.5 wt.%) that can be attributed to the grafted SiO_2_ nanoparticles. It was noticed, however, that the presence of a layer of non-conductive silica nanoparticles on the graphite surface in the hybrid filler does not constitute an obstacle to heat transport in the rubber matrix [30].

### 2.4. The Effect of the Dispersion of Additives in the Matrix on Thermal Resistance of Silicone Nanocomposites

Nanofillers are seen as replacements or auxiliary combinations of microfillers. They have a large surface area, which results in increased bonding strength and the improvement of various properties at filling levels much lower than microfillers. However, the basic problem with the use of nanofillers is their tendency to agglomerate as a consequence of their physical size and the forces acting between the nanofillers [52].

The analysis of the results of testing the thermal stability of silicone nanocomposites discussed in Section 2 indicates that an important factor whose influence is worth considering in detail is the method of introducing and dispersing nanofillers in the polysiloxane matrix. The use of an appropriate method of producing a nanocomposite makes it possible to obtain a uniform dispersion of nanofiller particles in the polymer matrix, which always has a beneficial effect on the properties of nanocomposites. This is especially clear when using nanofillers with a layered or plate structure, because they can create more heat conduction paths inside the polysiloxane matrix. This prevents the degradation of polysiloxane caused by excessive heat accumulation in local areas [47]. The basic methods tested are direct or ultrasonic dispersion, flocculation in solution, ultrasonic homogenization and wet jet milling. A comparison of the impact of these dispersion methods on the thermal stability of silicone nanocomposites is presented in Table 3.

Comparing the impact of different methods of dispersing nanofillers, it can be concluded that good dispersion is most important when using nanofillers with a large specific surface area. When using carbon nanotubes in an amount of 0.05 wt.%, it is possible to obtain a significant improvement in thermal stability when dispersing using the wet jet milling method. It is clearly visible that, even with such a small amount of nanofiller, it is possible to limit the loss of mass and the formation of cracks during thermal aging. The results obtained for polysiloxane nanocomposites indicate that the current practice of introducing carbon nanotubes in much larger quantities may also be changed in relation to other polymer matrices. This would significantly reduce the raw material costs of such nanocomposites. In the case of introducing graphene as a nanofiller in an amount of 0.8–1.5 phr, very good results expressed by an increase in the T_5_ temperature to 711 °C compared to 602 °C for rubber without fillers were obtained using ultrasonic dispersion. When considering the impact of the selected dispersion methods on the thermal stability of silicone nanocomposites, the viscosity of the polysiloxane should also be taken into account, which, as is known, can vary within a very wide range.

## 3. Analysis of Thermal Stability Mechanisms of Silicone Nanocomposites

The analysis of the literature data allows us to conclude that nanofillers influence the thermal stability of silicone nanocomposites, regardless of their structure and chemical composition. However, this impact varies. Therefore, it is advisable to compare the mechanisms of action of individual types of nanofillers in connection with their structure and chemical composition. According to the Tsagaropoulos model [57], the interfacial bond formed between the polymer molecular chain and the nanofiller limits the movement of the rubber molecular skeleton and hinders the formation of degradation products. Thus, it slows down the thermal degradation process.

CeO_2_ can limit the movement of polysiloxane chains interacting with graphene. The simultaneous use of these nanofillers, therefore, produces the desired synergy effect. Moreover, when polyphenylsiloxanes are used, there is a π-π conjugation effect between their phenyl groups and graphene with a ring structure. This generates a significant adsorption effect between graphene and polyphenylsiloxane, reducing the exposure of the polysiloxane chain ends. As a consequence, the formation of an intramolecular cyclic transition state is difficult, which increases the thermal stability of silicone rubber composites [58,59].

The introduction of a phenyl group, therefore, has a significant impact on the thermal stability of silicone rubber. This effect can be attributed to two aspects. First, the phenyl group is difficult to oxidize, which may also delay the oxidation of the methyl group. Secondly, the phenyl group in the silicone molecular chain can create a steric hindrance effect, which makes the -Si-O- segment difficult to form ring degradation products, and further improves the thermal stability of phenyl silicone rubber composites. Secondly, the reason for the significant improvement in the mechanical properties of silicone rubber nanocomposites containing less than 2 wt.% graphene as a nanofiller is the uniform dispersion of graphene in the rubber matrix and the formation of hydrogen bonds between graphene and polysiloxane chains [60]. However, the tensile strength and elongation at break of the nanocomposites decrease gradually with increasing graphene content. This is caused by excessive stiffening of the nanocomposite structure due to the formation of graphene-polysiloxane hydrogen bonds [61].

Based on detailed studies of polymer nanocomposites containing carbon nanotubes, it can be concluded that the surface of carbon nanotubes can scavenge radicals, which inhibits the chain reaction of polymer degradation [62]. This mechanism not only increases electrical conductivity and improves mechanical properties, but also has a significant impact on increasing thermal stability [28].

In the case of depolymerization of pure SR and SR/vinyl-GO nanocomposites, a radical mechanism was found to play a role [50]. Free radicals generated during depolymerization are adsorbed and shielded by vinyl-GO, which has a large specific surface area and two-dimensional geometry. It is hypothesized that these processes (i.e., adsorption and shielding) may be responsible for the improvement in Tmax1 with increasing vinyl-GO content. The Si-O framework can degrade via cyclic oligomers at higher temperature, which is probably due to the limitation of the cyclic transition state exerted by the cross-linked structures and the Karstedt catalyst. Furthermore, vinyl-GO is a critical factor that hinders the formation of cyclic oligomers [50].

The factors influencing thermal stability discussed in the review are presented in Figure 3, which schematically illustrates the influence of factors related to the polymer structure and the nature of nanofillers capable of various interactions with polysiloxane chains.

In polymer nanocomposites, an important factor is the nanofiller–polymer interactions, which, thanks to the synergistic effects, can significantly affect the properties of these nanocomposites, including thermal stability [63]. A significant increase in these interactions can be achieved by appropriate modification of the nanofiller surface. A frequently used method is modification with carbofunctional silanes, the so-called silane coupling agents, where the hydrophilic segment is attached to the nanoparticle with a surfactant, and the hydrophobic segment interacts with the base polymer [64]. Obtaining good modification results depends primarily on the proper selection of the silanizing agent, especially in terms of incorporation of appropriate functional groups depending on the type of nanofiller and polysiloxane [65,66]. The use of modified nanofillers reduces the surface energy and interfacial tension, contributing to the breakdown of agglomerates and the homogeneous introduction of nanoparticles into the structure of the polysiloxane nanocomposite.

It is worth adding that tools based on mathematical modelling are increasingly being used. A number of mathematical models have been developed, which constitute a good tool for the theoretical assessment of the thermal conductivity of polymer composites planned to be obtained. They take into account the size, shape, internal thermal conductivity and filler dispersion [67,68]. However, the vast majority of models are applicable to a single-filler composite [68,69,70,71,72]. However, taking into account the fact that the introduction of a single filler does not guarantee the achievement of the assumed parameters, models have also been developed to examine the mechanism of heat transfer in hybrid composites. One example is the Agrawal’s model [69], which allows for the theoretical determination of the thermal conductivity of polymer composites with hybrid fillers. The model was based on the law of minimum thermal resistance and the law of specific equivalent thermal conductivity. The effective thermal conductivity of polymers with particles of various conductive and non-conductive fillers was estimated. Based on the results obtained, it was found that the presence of two different fillers together contributes to the modification of the thermal conductivity values of such composites with hybrid fillers. Experimental efforts, which include fabricating different sets of composites and measuring their conductivity, confirm the suitability of the proposed model for predictive purposes. The theoretical and experimental values of effective thermal conductivity of various composites were comparable [69], whereas Vinod et al. [68] focused on determining the effective thermal conductivity of silicone rubber micro-nanocomposites, taking into account hybrid fillers (nano aluminium oxide and micro Alumina Trihydrate (ATH)) of unequal sizes and the agglomeration process. It was found, among others, that the volume fraction of aggregated nanoalumina fillers increases significantly with the increase in the weight percentage of nanoalumina fillers. Compared to the Agrawal model, the modified thermal conductivity model has a smaller error compared to the experimental data [68].

## 4. Conclusions

In this review, we focused our efforts on presenting selected solutions that allow for the management of the thermal stability of silicone nanocomposites. Inorganic nanoparticles, such as CeO_2_, TiO_2_, Al_2_O_3_, or carbon nanoparticles, as well as organic substances, under appropriate conditions (temperature, solvent, mixing, etc.) allow us to obtain a composite with increased parameters in terms of thermal stability compared to pure materials without fillers. Methods used in laboratories should be considered in terms of the possibility of their application on an industrial scale, taking into account the purpose of the modification, as well as the ease of the modification process or the cost of this process (stages, number of reagents, energy demand, e.g., in the case of using ultrasounds). Based on the analysed literature reports, it should be emphasized that the discussed polymer nanocomposites have a very large potential area of application in a number of innovative technological solutions due to their unique properties, especially very good thermal stability such as aerospace, automobile, construction, electric insulators and medical industries [73]. These nanocomposites could be also successfully used as strain sensors or as material of choice in electromagnetic interference shielding, as well as in microwave reflectivity applications [74]. The increasingly better performance achieved for silicone nanocomposites allows us to hope for an increase in their use and a significant share in the global silicone polymers market, which will increase from USD 18.59 billion in 2021 to USD 35.90 billion by 2030 at a Compound Annual Growth Rate (CAGR) of 6.4% during the forecast period. (According to a research report published by Spherical Insights & Consulting—Global Silicone Polymer Market Size, release date January 2023, Report ID SI1454).

## Figures and Tables

**Figure 1 materials-17-02016-f001:**
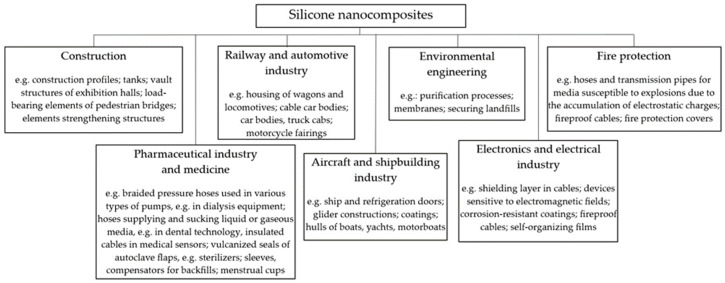
Examples of silicone nanocomposites in industrial and research applications.

**Figure 2 materials-17-02016-f002:**
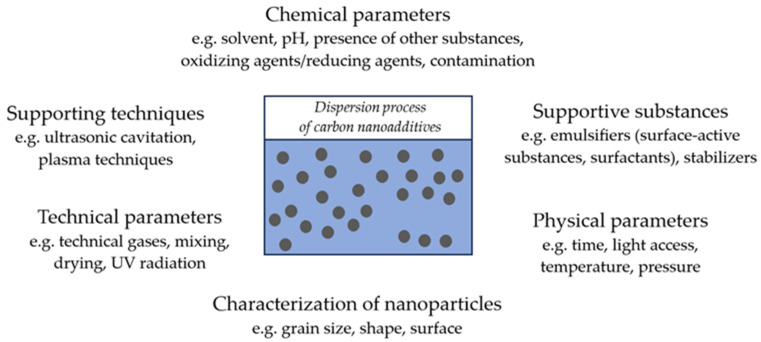
Factors influencing the dispersion of carbon nanoadditives.

**Figure 3 materials-17-02016-f003:**
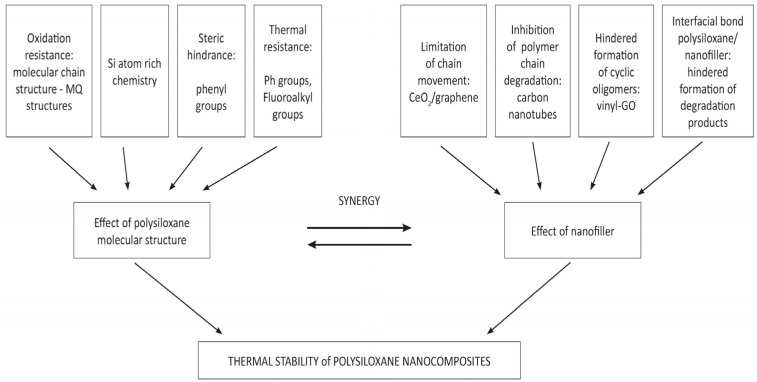
Factors affecting the thermal stability of silicone nanocomposites.

**Table 1 materials-17-02016-t001:** Comparison of the structural data of organosilicon and organic polymers.

Structure Element	Unit	Organosilicon Polymers	Organic Polymers
Interatomic distances	[nm]	Si-O 0.163	C-C 0.143
Angles between bonds	[deg]	Si-O-Si 143	C-O-C 112
Energy of bong formation	[kJ/mol]	Si-O 445 Si-C 306	C-O 358 C-C 318–352

**Table 2 materials-17-02016-t002:** The effect of R/Si and Ph/Me ratios on thermal resistance of organosilicon polymers.

Type of Organosilicon Polymer	R/Si	Ph/Me mol%	Solid Residue wt.% in Air 800 °C	T_5_	Ref
PDMS linear	2	0/100	39	175	[7,9]
Ph/Me polysiloxane linear	2	75/25	26.4	181	[10,11]
Me silicone resin	1.2	0/100	66.5	185	[7]
Me/Ph silicone resin	1.2	25/75	65.8	292	[7,12]
Me/Ph silicone resin	1.2	50/50	51.8	238	[7]
Me/Ph silicone resin	1.2	75/25	45.7	227	[7]
Me/Ph silicone resin	1.5	75/25	48.3	192	[7]
Ph silicone resin	1.2	100/0	40.5	242	[7]
MePh_2_SiO_1/2_- terminated MQ Me silicone resin	1.12		34.4	224.6	[13]
MQ Me/Ph silicone resin	0.76		74.4	424.5	[14]

PDMS—polydimethylsiloxane; Ph—phenyl; Me—methyl; MQ—M: mono-functional silicon–oxygen unit R_3_SiO_1/2_, Q: tetra-functional silicon–oxygen unit SiO_2_.

**Table 3 materials-17-02016-t003:** Comparison of the impact of selected dispersion methods on the thermal stability of silicone nanocomposites.

Polysiloxane	Nanofiller	Dispersion Method	Nanoparticle	Thermal Stability	Ref
Silicone rubber	Graphene, CeO_2_	Ultrasonically, 4 h w 50 °C	0.8–1.5 phr	T_5_ 711 °C compared to 602 °C for rubber without fillers	[53]
Silicone rubber	Graphene	Sonication in tetrahydrofuran; flocculationin ethanol	2.3 wt.%	Long-term heating does not change the extensibility	[54]
Silicone rubber	Magnesium hydroxide sulphate hydrate whisker	Introduction of nanofiller on rollers	7 phr	The residue after heating at 700 °C increases to 69% compared to 43% for the composite without the modifier	[55]
Silicone rubber	Carbon nanotubes	Comparison of 3 dispersion processes: mixing, sonication ultrasonic, wet jet milling	0.05 wt.%	Best nanotube dispersion after wet jet milling; after 7 days of heating at 280 °C, the nanocomposites retained their mechanical properties compared to composites without nanotubes, crumbling after 1 day of heating.	[56]

## Data Availability

Not applicable.

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
