# Peer review of "Silicone Nanocomposites with Enhanced Thermal Resistance: A Short Review"

_materials, 2024, doi:10.3390/ma17092016_

Round 1
Reviewer 1 Report
Comments and Suggestions for Authors
The work submitted for consideration in Materials is about the continuous advancements in technology that necessitate materials with enhanced long-term thermal resistance for modern devices. Heat-resistant polymer materials have been successful in such products, but there is a growing interest in exploring polymer nanocomposites for improved performance. Specifically, organosilicon polymers show promise due to their unique structural properties. This review examines literature on organosilicon polymer nanocomposites with heightened thermal resistance. It focuses on modifying siloxane molecular structures and incorporating inorganic additives or carbon nanomaterials to enhance thermal stability. Additionally, it delves into the dispersion of additives in the polymer matrix and the potential for permanently introducing nanofillers through reactive groups. The thermal stability mechanism of these nanocomposites is also investigated.
From a piece of text in red I supposed this is a revision of the first submission, however the new piece of text is not complete: “Al2O3 nanoparticles were also used by Lorenzo [28]. Aluminum oxide nanoparticles were combined with SiO2 nanoparticles to obtain the SiO2@Al2O3 nanofiller or to , which was used to fill SBR and BR.”
Other kinds of silicon based materials like halloysites are missing apart from the red piece of text: DOI:10.1016/j.jpcs.2021.109949; DOI:10.1016/j.jiec.2021.02.031;
In the references the journals abbreviations are missing in for example ref. 16, revise if the abbreviation “Polym.” exists and actually, revise all the references because there are other errors like in refs. 19, 20…
The discussion on the differences because of the temperature and specially about the stability is remarkable. Only I would request to add some adjective like “insignificant” difference of 9 or 15ºC that in some cases appear, which is rather small.
However, I recommend the current review for publication since it will be fore sure cited, thus the impact is quite ensured since the next studies on silicon based polymers will use it for sure.
Author Response
Response to Reviewer 1 Comments
General comment
The work submitted for consideration in Materials is about the continuous advancements in technology that necessitate materials with enhanced long-term thermal resistance for modern devices. Heat-resistant polymer materials have been successful in such products, but there is a growing interest in exploring polymer nanocomposites for improved performance. Specifically, organosilicon polymers show promise due to their unique structural properties. This review examines literature on organosilicon polymer nanocomposites with heightened thermal resistance. It focuses on modifying siloxane molecular structures and incorporating inorganic additives or carbon nanomaterials to enhance thermal stability. Additionally, it delves into the dispersion of additives in the polymer matrix and the potential for permanently introducing nanofillers through reactive groups. The thermal stability mechanism of these nanocomposites is also investigated.
From a piece of text in red I supposed this is a revision of the first submission, however the new piece of text is not complete: “Al2O3 nanoparticles were also used by Lorenzo [28]. Aluminum oxide nanoparticles were combined with SiO2 nanoparticles to obtain the SiO2@Al2O3 nanofiller or to , which was used to fill SBR and BR.”
Other kinds of silicon based materials like halloysites are missing apart from the red piece of text: DOI:10.1016/j.jpcs.2021.109949; DOI:10.1016/j.jiec.2021.02.031; https://doi.org/10.1016/j.porgcoat.2019.01.029 POC 2019, vol 129, 357-365 Well-cured silicone/halloysite nanotubes + Influence of halloysite nanotubes on mechanical and swelling properties of silicone rubber compound
In the references the journals abbreviations are missing in for example ref. 16, revise if the abbreviation “Polym.” exists and actually, revise all the references because there are other errors like in refs. 19, 20…
The discussion on the differences because of the temperature and specially about the stability is remarkable. Only I would request to add some adjective like “insignificant” difference of 9 or 15ºC that in some cases appear, which is rather small.
However, I recommend the current review for publication since it will be fore sure cited, thus the impact is quite ensured since the next studies on silicon based polymers will use it for sure.
We would like to thank the reviewer for his careful and insightful reading of this manuscript and for his thoughtful comments and constructive suggestions that will help to improve the quality of this manuscript. With respect to the recommendations, we have made appropriate corrections and additions. We hope that the article in its current version will meet the Reviewer's expectations in this respect.

Reviewer 2 Report
Comments and Suggestions for Authors
The authors reviewed the results of the most important literature reports regarding organosilicon polymer nanocomposites with increased thermal resistance. The modification methods of silicone nanocomposites focused on the increase of their thermal resistance related to the modification of siloxane molecular structure and by making nanocomposites using inorganic additives or carbon nanomaterials. The influence of the dispersion of additives in the polymer matrix on the thermal resistance of silicone nanocomposites, modifying the polymer matrix, permanently introducing nanofillers thanks to various reactive groups and the thermal stability mechanism of these nanocomposites were analysed.
The reviewers make the following comments.
1. Please add references from the last 3 years.
2. Modify the relevant content according to the added literature.
Author Response
Response to Reviewer 2 Comments
General comment
The authors reviewed the results of the most important literature reports regarding organosilicon polymer nanocomposites with increased thermal resistance. The modification methods of silicone nanocomposites focused on the increase of their thermal resistance related to the modification of siloxane molecular structure and by making nanocomposites using inorganic additives or carbon nanomaterials. The influence of the dispersion of additives in the polymer matrix on the thermal resistance of silicone nanocomposites, modifying the polymer matrix, permanently introducing nanofillers thanks to various reactive groups and the thermal stability mechanism of these nanocomposites were analysed.
The reviewers make the following comments.
- Please add references from the last 3 years.
- Modify the relevant content according to the added literature.
We would like to thank the Reviewer for careful and thorough reading of this manuscript and for the thoughtful comments and constructive suggestions, which help to improve the quality of this manuscript. With respect to the recommendations, we have made appropriate corrections and additions. We have added literature from recent years and supplemented the text. We hope that the article in its current version will meet the Reviewer's expectations in this respect.

Reviewer 3 Report
Comments and Suggestions for Authors
Dear authors
I have overall enjoyed article Reading. The topic discussed by the authors is interesting to the readers, and in general the article is well written. I list below some major and minor changes that must be addressed before further article processing
Major changes
What were the keywords used during the article search? Please also specify the years considered and the search motor (scopus, WOS or similar)? How many results were found, how were they filtered?
In the introduction section, please provide some reference articles that report current usage of silicone nanocomposites in industrial and research applications
Minor corrections
Line 259: what do you mean with “loss temperature”?. Please consider sentence rewriting for a better comprehension.
Comments on the Quality of English LanguageLine 264. This paragraph was written using past tense, however the sentence “this system allows for …” is written in present tense. Please consider using a single tense for every paragraph.
Author Response
Response to Reviewer 3 Comments
General comment
Dear authors
I have overall enjoyed article Reading. The topic discussed by the authors is interesting to the readers, and in general the article is well written. I list below some major and minor changes that must be addressed before further article processing
Major changes
What were the keywords used during the article search? Please also specify the years considered and the search motor (scopus, WOS or similar)? How many results were found, how were they filtered?
We would like to thank the Reviewer for his comment. We have entered the relevant information into the manuscript. We would like to add that we used the keywords: silicone nanocomposites, thermal stability, nanoadditives, i.e. appropriately to the article. As a result of the search, we received 28 results. However, the vast majority of patents focused on mechanical properties, which did not fit the scope of the presented article.
In the introduction section, please provide some reference articles that report current usage of silicone nanocomposites in industrial and research applications
We would like to thank the Reviewer for his comment. As recommended, we have added examples of silicone nanocomposites in industrial and research applications to the article in the form of Fig. 1.
Minor corrections
Line 259: what do you mean with “loss temperature”?. Please consider sentence rewriting for a better comprehension.
Line 264. This paragraph was written using past tense, however the sentence “this system allows for …” is written in present tense. Please consider using a single tense for every paragraph.
We would like to thank the Reviewer for comments. We have introduced appropriate corrections.
We hope that the article in its current version will meet the reviewer's expectations in this regard.

Reviewer 4 Report
Comments and Suggestions for Authors
This manuscript gives a review on the enhanced thermal resistance behavior of silicone nanocomposites. The manuscript is well organized and minor revisions are requested. My comments are below:
1. Figures are suggested.
2. The thermal resistance mechanism of silicone nanocomposites should be explored.
3. Too much contents of carbon nanoparticles are stated instead of silicon nanocomposites.
Author Response
Response to Reviewer 4 Comments
General comment
This manuscript gives a review on the enhanced thermal resistance behavior of silicone nanocomposites. The manuscript is well organized and minor revisions are requested.
We would like to thank the reviewer for his careful and insightful reading of this manuscript and for his thoughtful comments and constructive suggestions that will help to improve the quality of this manuscript.
My comments are below:
- Figures are suggested.
We would like to thank the reviewer for his comment. As recommended, we added figures to the manuscript.
- The thermal resistance mechanism of silicone nanocomposites should be explored.
We would like to thank the reviewer for his comment. We added figure to the chapter 3 of the manuscript as well as the description concerning factors influencing thermal stability discussed in the review.
- Too much contents of carbon nanoparticles are stated instead of silicon nanocomposites.
We have dedicated one subchapter to presenting nanoadditives in the form of various forms of carbon for silicones, as it is one of the most frequently used additives. Therefore, we believe that leaving the chapter in this form is appropriate for the content of the article. But in response to this comment, we have also added information to other chapters to reduce the differences.
We hope that the article in its current version will meet the Reviewer's expectations in this respect.

Reviewer 5 Report
Comments and Suggestions for Authors
This manuscript deals with the topic of thermal resistance in silicon nanocomposites, with focus on the structure, properties, modification provoked by the filler into the matrices and thermal stability mechanisms which are involved in. The article has potential but I consider it could be accepted after some improvements and major revisions.
1- Literature should be updated. Only a few papers of the last few years are included.
2- I believe the manuscript could be considered a mini-review, so I suggest including the word “short” in the title.
3- Inclusion of some figures and more tables is really important in this article type. I suggest to the authors the incorporation of some schemes/figures to describe better the use of polysiloxanes, for example.
4- A list of abbreviations could be added to facilitate the readership understanding.
5- Introduction. In some parts, references are missing. Also, this section should be improved and highlight the weakness and importance of different materials and approaches.
6- Surface modification processes should be explained deeply.
7- Line 309. Consider a section, no chapter.
8- Line 318. Please, expand the idea and describe it in detail.
9- Section 2.3. Is desirable the incorporation of any table or scheme.
10- Discussions and more comprehensive analysis of literature is needed throughout the whole manuscript.
11- Conclusions. This section seems a little poor; I recommend re-writing and also give the perspectives and personal views.
Author Response
Response to Reviewer 5 Comments
General comment
This manuscript deals with the topic of thermal resistance in silicon nanocomposites, with focus on the structure, properties, modification provoked by the filler into the matrices and thermal stability mechanisms which are involved in. The article has potential but I consider it could be accepted after some improvements and major revisions.
We would like to thank the Reviewer for his careful and insightful reading of this manuscript and for his thoughtful comments and constructive suggestions that will help to improve the quality of this manuscript.
Below is information about the corrections and additions introduced:
1- Literature should be updated. Only a few papers of the last few years are included.
We would like to thank the Reviewer for this comment. We have added literature from recent years.
2- I believe the manuscript could be considered a mini-review, so I suggest including the word “short” in the title.
We would like to thank the Reviewer for this comment. As recommended, we added information to the title.
3- Inclusion of some figures and more tables is really important in this article type. I suggest to the authors the incorporation of some schemes/figures to describe better the use of polysiloxanes, for example.
We would like to thank the Reviewer for this comment. As recommended, we added figures (Fig. 1, Fig. 2) to the manuscript.
4- A list of abbreviations could be added to facilitate the readership understanding.
We would like to thank the Reviewer for this comment. As recommended, we have added a list of abbreviations.
5- Introduction. In some parts, references are missing. Also, this section should be improved and highlight the weakness and importance of different materials and approaches.
We would like to thank the Reviewer for this comment. As recommended, we have added supplements and literature references.
6- Surface modification processes should be explained deeply.
We would like to thank the Reviewer for this comment. As recommended, we have added the description concerning surface modification and literature references.
7- Line 309. Consider a section, no chapter.
We would like to thank the Reviewer for this comment.We have corrected the entry.
8- Line 318. Please, expand the idea and describe it in detail.
We would like to thank the Reviewer for this comment. As recommended we have added the detailed description comparing the impact of different methods of dispersing nanofillers.
9- Section 2.3. Is desirable the incorporation of any table or scheme.
We would like to thank the Reviewer for this comment. As recommended, we have added the appropriate scheme.
10- Discussions and more comprehensive analysis of literature is needed throughout the whole manuscript.
We would like to thank the Reviewer for this comment. In accordance with the recommendations, we have introduced additional additions.
11- Conclusions. This section seems a little poor; I recommend re-writing and also give the perspectives and personal views.
We would like to thank the Reviewer for this comment. In accordance with the recommendations, we have introduced additional additions.
We hope that all done corrections and additions improved the quality of the manuscript and the article in its current version will meet the reviewer's expectations in this regard.

Round 2
Reviewer 3 Report
Comments and Suggestions for Authors
Mandatory changes have been replied one-by-one. The paper can be published in present form
Reviewer 5 Report
Comments and Suggestions for Authors
The authors made substantial improvements to the manuscript.